# Impact of Progressive Site-Directed Therapy in Oligometastatic Castration-Resistant Prostate Cancer on Subsequent Treatment Response

**DOI:** 10.3390/cancers14030567

**Published:** 2022-01-23

**Authors:** Soichiro Yoshida, Taro Takahara, Yuki Arita, Kazuma Toda, Koichiro Kimura, Hajime Tanaka, Minato Yokoyama, Yoh Matsuoka, Ryoichi Yoshimura, Yasuhisa Fujii

**Affiliations:** 1Department of Urology, Tokyo Medical and Dental University, Tokyo 113-8510, Japan; hjtauro@tmd.ac.jp (H.T.); mntykym.uro@tmd.ac.jp (M.Y.); yoh-m.uro@tmd.ac.jp (Y.M.); y-fujii.uro@tmd.ac.jp (Y.F.); 2Department of Biomedical Engineering, School of Engineering, Tokai University, Isehara 259-1193, Japan; tarorin@gmail.com; 3Department of Radiology, Yaesu Clinic, Advanced Imaging Center, Tokyo 103-0027, Japan; 4Departments of Diagnostic Radiology, Keio University School of Medicine, Tokyo 160-8582, Japan; yukiarita1113@gmail.com; 5Department of Radiation Therapeutics and Oncology, Tokyo Medical and Dental University, Tokyo 113-8510, Japan; tdmrad@tmd.ac.jp (K.T.); ysmrmrad@tmd.ac.jp (R.Y.); 6Department of Diagnostic Radiology, Tokyo Medical and Dental University, Tokyo 113-8510, Japan; kmrdrnm@gmail.com

**Keywords:** diffusion magnetic resonance imaging, whole body imaging, radiotherapy, prostatic neoplasms, castration-resistant, neoplasm metastasis

## Abstract

**Simple Summary:**

Local treatment for oligometastatic hormone-naive prostate cancer has been shown to be effective in phase II trials. As for the efficacy of targeted therapy for oligometastatic castration-resistant prostate cancer, the results of the phase trial are not yet available, but the number of reports showing efficacy by retrospective analysis is increasing. Progressive site-directed therapy has been shown to delay the next intervention and prolong progression-free survival, but its impact on subsequent treatment efficacy and contribution to overall survival has not been reported. The purpose of this retrospective study is to evaluate the impact of progressive site-directed therapy for oligometastatic castration-resistant prostate cancer on the subsequent treatment outcomes. We found that progressive site-directed therapy was associated with better response to subsequent androgen receptor axis-targeted drugs and better overall survival. Progressive site-directed therapy for oligometastatic castration-resistant prostate cancer may improve subsequent oncological outcomes.

**Abstract:**

The purpose of this study was to evaluate the impact of progressive site-directed therapy (PSDT) for oligometastatic castration-resistant prostate cancer (OM-CRPC) on the efficacy of subsequent androgen receptor axis-targeted (ARAT) drugs, and to demonstrate the possibility of prolonging overall survival (OS). We performed a retrospective analysis of 15 OM-CRPC patients who underwent PSDT and subsequently received first-line ARAT drugs (PSDT group) and 13 OM-CRPC patients who were treated with first-line ARAT drugs without PSDT (non-PSDT group). PSDT was performed with the intention of treating all progressing sites detected by whole-body diffusion-weighted MRI with radiotherapy. Thirteen patients (86.7%) treated with PSDT had a decrease in PSA levels, which was at least 50% in 10 (66.7%) patients. The median PSA progression-free survival (PFS) for PSDT was 7.4 months. The median PSA-PFS for ARAT was 27.2 months in patients in the PSDT group and 11.7 months in the non-PSDT group, with a significant difference between the two groups (hazard ratio [HR], 0.28; *p* = 0.010). The median OS was not reached in the PSDT group and was significantly longer than 44.5 months in the non-PSDT group (HR, 0.11; *p* = 0.014). In OM-CRPC, PSDT may improve the efficacy of subsequent ARAT and OS.

## 1. Introduction

In oligometastatic cancer, a disease with limited number of metastases, local treatment of metastases was proposed to be effective based on the idea that the oligometastatic cancer has a similar biological behavior as localized disease [1,2,3,4,5]. A prospective, randomized, controlled phase II trial of oligometastatic hormone-naive prostate cancer showed that local treatment of metastatic disease delayed the commencement of androgen deprivation therapy and prolonged progression-free survival (PFS) [6,7,8]. Recently, the frequency of driver mutations including TP53, WNT, and cell cycle genes in metastatic hormone-naive prostate cancer has been shown to correlate directly with increased metastatic burden, consistent with the spectral theory of metastasis, including the concept of oligometastatic prostate cancer [9]. 

As the benefits of local treatment for metastatic hormone-naive prostate cancer become more established, there is growing interest in the implications of a multidisciplinary treatment option that combines systemic therapy with progressive site-directed therapy (PSDT) for castration-resistant prostate cancer (CRPC) in which only a few lesions have progressed (oligometastatic CRPC; OM-CRPC) [10,11,12,13]. Although there are no reported results of randomized controlled trials on OM-CRPC, several retrospective studies have reported favorable prostate-specific antigen (PSA) responses and delayed clinical progression with PSDT, suggesting that addition of targeting treatment for progressing lesions can be an effective treatment option for OM-CRPC, including our reports [14,15,16,17,18,19,20]. However, little is known about the effect of PSDT on the efficacy of subsequent androgen receptor axis-targeted (ARAT) drugs [21].

Considering that progressing lesion in OM-CRPC includes cancer cell clones that are resistant to systemic therapy being administered, PSDT may be effective in reducing these treatment-resistant clones and thus contributing to improved efficacy of subsequent systemic therapy. Although our results are from a small number of cohorts with short follow up after PSDT, we have reported a trend towards better efficacy of ARAT drugs administered to patients with OM-CRPC after undergoing PSDT compared with those who did not undergo PSDT [21]. The purpose of this study was to evaluate the impact of PSDT for OM-CRPC on the efficacy of subsequent ARAT drugs, and to demonstrate the possibility of prolonging overall survival (OS) in a cohort with an additional number of patients, and an extended follow-up period.

## 2. Materials and Methods

This study included 71 patients diagnosed with OM-CRPC by whole-body diffusion-weighted MRI (WB-DWI) who underwent diffusion-weighted whole-body imaging with background body signal suppression (DWIBS) before starting a new line of anticancer therapy at a single institution between 2014 and 2020. Seventeen patients with prior ARAT treatment were excluded from the study. Twenty-six were also excluded from the analysis because ARAT was not introduced as the first systemic treatment after the diagnosis of OM-CRPC.

CRPC patients were defined as having serum testosterone < 50 ng/dL, and biochemical or radiological progression according to the European association of urology guideline [22]. The details of WB-DWI have been previously reported [23,24,25]. OM-CRPC patients were defined as having 3 or less progressive lesions assessed based on WB-DWI; up to 2 progressive metastatic lesions were allowed for those with a progressive lesion in the prostate.

Of the OM-CRPC patients, we evaluated the treatment outcomes in patients who underwent PSDT and were subsequently treated with first-line ARAT drugs (PSDT group) and those who were treated with first-line ARAT drugs without PSDT (non-PSDT group). 

PSDT was performed with the intention of treating all progressing lesions detected by WB-DWI with radiotherapy. Prostate/lymph node metastases were treated with 60–78 Gy (2 Gy per fraction), and bone metastases were treated with 30–39 Gy (2–3 Gy per fraction) external beam radiotherapy. Systemic treatment was unchanged when PSDT was performed according to the institutional treatment protocol; after 2019, PSDT was combined with radium-223 and bone-modifying agent in cases where progressing metastatic lesions was confined to bone. Patients were typically visited every 1–3 months and based on an evaluation that included a physical examination, PSA, and imaging studies, changes in systemic therapy and repeated PSDTs were performed by their respective physicians.

Differences in the characteristics of patients in the PSDT and non-PSDT groups were compared using the Wilcoxon test for continuous variables, and Fisher’s exact test for nominal variables. The PSA-PFS for PSDT was calculated as the time from the start of radiotherapy to PSA progression (2 ng/mL increase from the PSA nadir), and patients who did not show a decrease in PSA were considered to have PSA progression at time point 0. PSA-PFS for ARAT was defined as the date when an increase of 25% or more from the nadir and an absolute increase in PSA of at least 2 ng/mL are confirmed by a second measurement obtained at least 3 weeks later. PFS and OS were estimated by the Kaplan–Meier method and compared among subgroups using the log-rank test. To identify variables associated with PSA-PFS and OS, Cox regression analysis was performed using clinical factors at the time of WB-DWI examination—notwithstanding if PSDT was performed—and ARAT drugs administered as analysis factors. Early adverse effects of PSDT treatment were assessed using the Radiation Therapy Oncology Group (RTOG) scale [26]. All statistical analyses were performed using JMP^®^ 14 (SAS Institute Inc., Cary, NC, USA), and *p* < 0.05 was considered significant.

## 3. Results

Of the 28 patients included in the analysis, 15 patients were in the PSDT group and 13 patients were in the non-PSDT group. The median follow-up time for the PSDT and non-PSDT groups was 38.5 (range, 7.2–72.9) months and 33.3 (range, 8.4–55.4) months, respectively. The characteristics of the PSDT and the non-PSDT groups at WB-DWI examination are shown in Table 1. There were no significant differences between the PSDT and non-PSDT groups in age, PSA level, PSA doubling time, number of progressing lesions, or number of treatment lines received at the time of WB-DWI examination. Eleven patients (73.3%) in the PSDT group and four patients (30.8%) in the non-PSDT group had a history of definitive treatment for the prostate, which was significantly more frequent in the PSDT group (*p* = 0.030). Of the PSDT group, all 11 patients with a history of definitive prostate treatment had no progressing disease within the prostate and did not include the prostate in their PSDT, while two of the remaining four patients with no prior definitive prostate treatment had progressing disease in the prostate along with metastatic disease and included the prostate in their PSDT.

The median number of progressing lesions in the PSDT group was 1 (range, 1–3), and the median time from WB-DWI to PSDT was 1.6 (range, 0.67 to 2.5) months. The most frequent target organ of PSDT was bone alone (*n* = 9), followed by lymph nodes alone (*n* = 3). Thirteen patients (86.7%) treated with PSDT had a decrease in PSA levels, which was at least 50% in 10 (66.7%) patients. PSDT was combined with radium-223 and bone-modifying agent in one patient. The median time to PSA progression was 7.4 months (range, 1.9–25.5 months, Figure 1), and the median time to initiation of ARAT was 10.5 (range, 6.21–22.5) months. Early toxicity of PSDT of Grade 2 or higher was observed in one patient (4%) (proctitis, Radiation Therapy Oncology Group Grade 2). 

As the first ARAT, eight patients (53.3%) received enzalutamide and seven (46.7%) received abiraterone in the PSDT group, and nine patients (69.2%) received enzalutamide and four (30.8%) received abiraterone in the non-PSDT group. The median number of lines of systemic treatment administered after the diagnosis of OM-CRPC was one (1–4) in the PSDT group and three (1–7) in the non-PSDT group, significantly more in the non-PSDT group (*p* = 0.0085).

The median PSA-PFS to ARAT was 27.2 months in patients in the PSDT group and 11.7 months in the non-PSDT group, with a significant difference between the two groups (hazard ratio [HR], 0.28; 95% confidence interval [CI], 0.10–0.74; *p* = 0.010, Figure 2). Of the factors evaluated, only PSDT implementation was associated with PSA-PFS to ARAT drug (Table 2). Similarly, the median PSA-PFS from diagnosis of OM-CRPC to PSA progression to ARAT was not reached in the PSDT group, significantly longer than 16.8 months in the non-PSDT group (HR, 0.11; 95% CI, 0.03–0.33; *p* < 0.0001, Figure 3). OS from the diagnosis of OM-CRPC was not reached in the PSDT group and was significantly longer than 44.5 months in the non-PSDT group (HR, 0.11; 95% CI, 0.006–0.67; *p* = 0.014, Figure 4). Of the variables evaluated, only the presence of PSDT was associated with OS (Table 3).

## 4. Discussion

This is the first study to show that PSDT has a positive impact on the therapeutic response to subsequent ARAT drug and the OS in OM-CRPC, together with the extension of time to next intervention (TTNI) due to the therapeutic effect of PSDT. PSDT for OM-CRPC is recommended because of the potential to improve the subsequent clinical course and mild adverse effects.

In progressing metastatic CRPC, there is a mixture of responsive and non-responsive lesions when the disease becomes drug resistant, confirming the heterogeneity of lesion response to treatment in metastatic CRPC [27,28]. Clones resistant to treatment contribute to the development of new macroscopic metastases. Given that the progressive lesions in OM-CRPC include clones that are resistant to current systemic therapy, PSDT may be effective in reducing these treatment-resistant clones and may contribute to the improved efficacy of subsequent systemic therapy [29].

Several retrospective cohort studies have investigated the therapeutic effect of PSDT to OM-CRPC and have reported promising results [14,15,16,17,18,19]. From an analysis of data viewed retrospectively on the outcomes of 68 patients with OM-CRPC, defined as five or fewer progressing lesions, treated with stereotactic ablative radiotherapy (SABR), Deek et al. reported that the time to PSA recurrence, TTNI, and median distant metastasis-free survival (DMFS) after PSDT were 9.7, 15.6, and 10.8 months, respectively [14]. Furthermore, in an analysis excluding those who did not change the systematic treatment at the time of SABR, there was a trend toward improved PSA-PFS (19.8 vs. 4.2 months, *p* = 0.05), DMFS (23.6 vs. 8.9 months, *p* = 0.007), and TTNI (20.3 vs. 8.8 months, *p* = 0.079) in OM-CRPC who underwent SABR in addition to change in systemic therapy compared with OM-CRPC who underwent change in systemic therapy alone and no SABR, indicating that the combination of SABR and change in systemic therapy for OM-CRPC has the potential to improve oncologic outcome [14].

The effect of metastasis-directed therapy on OS prolongation has been shown in a randomized, open-label, phase II trial, SABR-COMET (NCT01446744), in oligometastatic cancers of various types, of which prostate cancer accounts for 16%. SABR-COMET showed that adding SABR to consolidate all metastatic sites in addition to standard treatment improved OS (median 50 vs. 28 months, HR 0.47; *p* = 0.006) for patients with a controlled primary tumor and five or fewer metastatic lesions [30]. The present study adds important insights into the impact of PSDT on the clinical course of treatment after PSDT for OM-CRPC. The results of this study suggest that PSDT may be able to improve the efficacy of subsequent systemic therapies after progression to PSDT, which may lead to a longer OS. Furthermore, this approach is expected to prevent the appearance of symptoms caused by local tumor progression and maintain quality of life, highlighting the benefits of an integrated approach including PSDT. 

In summary, PSDT for oligoprogressive lesions is effective and there may be advantages to performing PSDT prior to initiating ARAT as the next systemic therapy. To confirm these promising results, the results of prospective, randomized, controlled trials examining the efficacy of PSDT for OM-CRPC are awaited. Randomized controlled trials evaluating the efficacy of PSDT for OM-CRPC are currently underway, including the ARTO trial (NCT03449719) to evaluate the effect of adding SBRT to abiraterone acetate, the DECREASE trial (NCT04319783) to evaluate the effect of adding SBRT to dalolutamide, and the FORCE trial (NCT03556904) to evaluate the effect of adding SBRT to standard therapy.

There are several limitations to this study. First, this study has a small number of cases and is retrospective in nature. In this study, the diagnosis of OM-CRPC was made by WB-DWI using a uniform protocol in both PSDT and non-PSDT groups, and although there were no significant differences in patient characteristics between the PSDT and non-PSDT groups, other than the history of definitive therapy for prostate cancer, there is an inherent potential for bias in the selection of patients for PSDT. Second, in order to evaluate the contribution of PSDT to the subsequent course of treatment for OM-CRPC, this study compared a cohort of patients in whom PSDT was not performed at the time of diagnosis of oligoprogression and only ARAT initiation was performed, with a cohort of patients in whom PSDT was followed by ARAT initiation. Therefore, of the patients who underwent PSDT, only those with disease progression after PSDT were included in the analysis, while those with continued therapeutic response to PSDT were excluded. This led to underestimation of the therapeutic effect of PSDT. Third, SBRT, which has been shown to be effective in treating metastatic disease in recent years, was not used in the PSDTs in this study, which may also underestimate the therapeutic effect of PSDTs. Fourth, the non-PSDT group included one case each of liver and lug metastasis, which generally have a poor prognosis. This patient selection bias may have affected the results of this study.

## 5. Conclusions

In OM-CRPC, PSDT may result in a better response to subsequent ARAT and better OS. Future prospective studies are necessary to confirm these results.

## Figures and Tables

**Figure 1 cancers-14-00567-f001:**
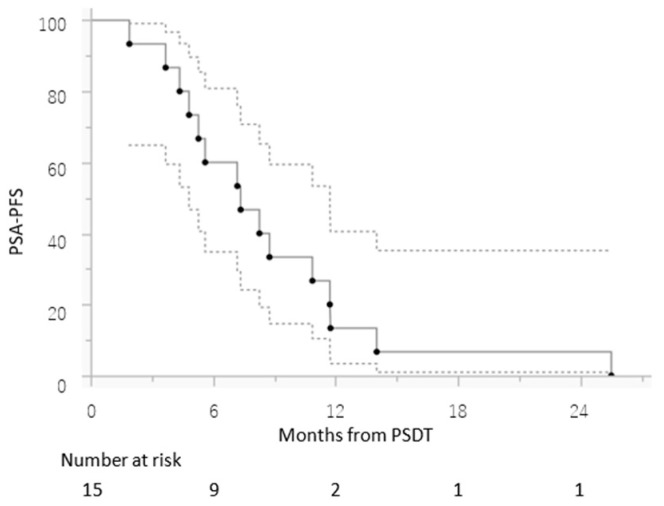
PSA progression-free survival (PSA-PFS) for patients with oligometastatic castration-resistant prostate cancer treated with progressive site-directed therapy (PSDT). The 95% intervals calculated for each time point are shown as dashed lines.

**Figure 2 cancers-14-00567-f002:**
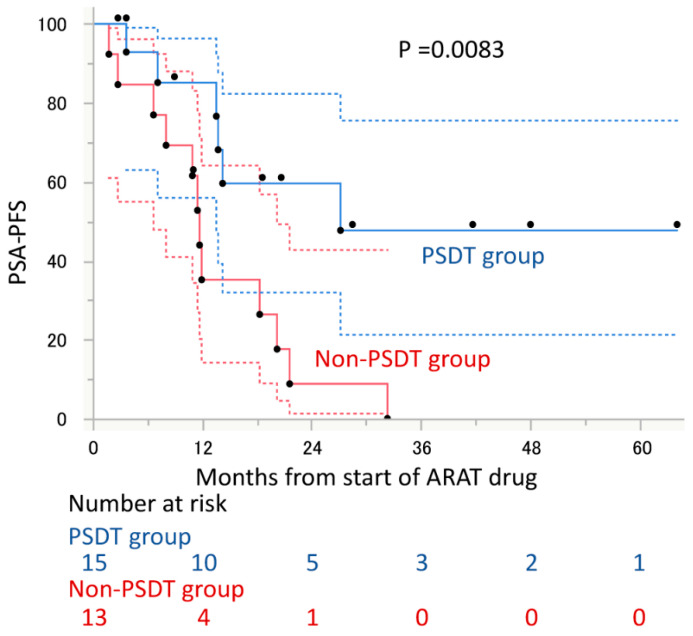
PSA progression-free survival (PSA-PFS) from the start of androgen receptor axis-targeted (ARAT) drugs in oligometastatic castration-resistant prostate cancer patients stratified by performance of progressive site-directed therapy (PSDT). The 95% intervals calculated for each time point are shown as dashed lines.

**Figure 3 cancers-14-00567-f003:**
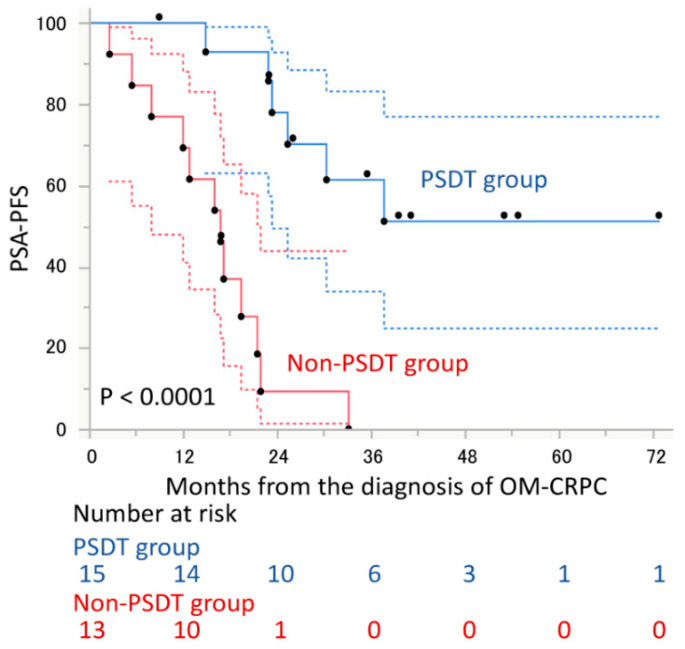
PSA progression-free survival (PSA-PFS) from the diagnosis of oligometastatic castration-resistant prostate cancer (OM-CRPC) stratified by performance of progressive site-directed therapy (PSDT). The 95% intervals calculated for each time point are shown as dashed lines.

**Figure 4 cancers-14-00567-f004:**
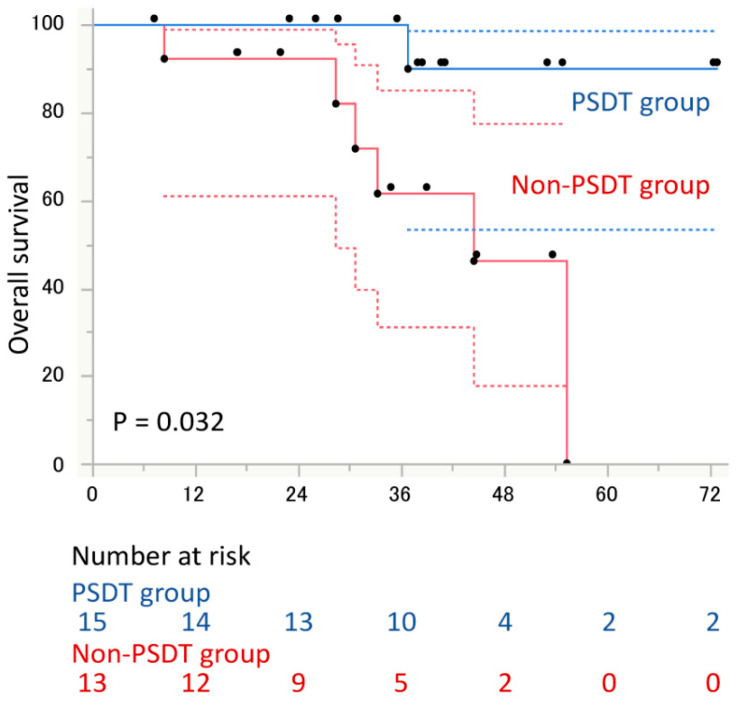
Overall survival from the diagnosis of oligometastatic castration-resistant prostate cancer (OM-CRPC) stratified by performance of progressive site-directed therapy (PSDT). The 95% intervals calculated for each time point are shown as dashed lines.

**Table 1 cancers-14-00567-t001:** Patient and treatment characteristics of oligometastatic castration-resistant prostate cancer patients treated with an androgen receptor-axis-targeted drug.

	PSDT ^†^ Group (*n* = 15)	Non-PSDT Group (*n* = 13)	*p* Value
Age (year), median (range)	76 (68–87)	77 (69–87)	0.87
PSA * at the diagnosis of OM-CRPC ^¶^ (ng/ml), median (range)	4.96 (0.22–32.0)	4.33 (0.70–26.8)	0.84
PSA-DT ^‡^ at the diagnosis of OM-CRPC (month), median (range)	3.1 (1.6–7.0)	3.8 (0.9–18.1)	0.12
No. of received treatment lines, median (range) before WB-DWI	2 (1–4)	2 (1–4)	0.98
Received treatment lines before the diagnosis of OM-CRPC			
Surgical orchiectomy/Leuprorelin/Goserelin/Degarelix	15 (100%)	13 (100%)	
Bicalutamide	12 (80.0%)	11 (84.6%)	
Estramustine phosphate	1 (6.7%)	4 (30.8%)	
Flutamide	1 (6.7%)	3 (23.1%)	
Docetaxel	1 (6.7%)	1 (7.7%)	
History of local treatment	11 (73.3%)	4 (30.8%)	0.030
No. of progressive sites	1 (1–3)	2 (1–3)	0.37
Location of progressive sites			
Prostate + bone	2 (15.4%)	3 (20.0%)	
Bone	9 (69.2%)	4 (26.7%)	
Prostate + lymph node	1 (7.7%)	2 (13.3%)	
Lymph node	3 (23.1%)	1 (6.7%)	
Bone + lymph node	0	1 (6.7%)	
Lung	0	1 (6.7%)	
Prostate + Liver + lymph node	0	1 (6.7%)	
Received 1st ARAT ^§^ after the diagnosis of OM-CRPC			
Enzalutamide	8 (53.3%)	9 (69.2%)	0.32
Abiraterone	7 (46.7%)	4 (30.8%)	
No. of received systemic treatment lines after the diagnosis of OM-CRPC, median (range)	1 (1–4)	3 (1–7)	0.0085
Received systemic treatment lines after the diagnosis of OM-CRPC			
Enzalutamide	10 (66.7%)	11 (84.6%)	
Abiraterone	8 (53.3%)	8 (61.5%)	
Docetaxel	2 (13.3%)	9 (69.2%)	
Cabazitaxel	3 (20.0%)	4 (30.1%)	
Radiuim-223	2 (13.3%)	3 (23.1%)	
Oraparib	0	1 (7.7%)	

PSDT ^†^, progressive site-directed therapy; PSA *, prostate-specific antigen; PSA-DT ^‡^, PSA doubling time; OM-CPRC ^¶^, oligometastatic castration-resistant prostate cancer; ARAT ^§^, androgen receptor-axis-targeted.

**Table 2 cancers-14-00567-t002:** Cox hazard model to identify factors predictive of time to PSA progression for an androgen receptor-axis-targeted drug.

Variable		N	Univariate	
	Category	Patients	HR ^††^ (95% CI ^§^)	*p* Value
Age at the diagnosis of OM-CRPC ^#^			1.05 (0.97–1.05)	0.24
PSA * at the diagnosis of OM-CRPC			1.03 (0.97–1.09)	0.24
PSA-DT ^‡^ at the diagnosis of OM-CRPC			1.07 (0.93–1.20)	0.27
History of definitive therapy for prostate cancer	no vs. yes	13/15	0.91 (0.36–2.41)	0.51
PSDT ^†^	no vs. yes	13/15	0.28 (0.10–0.74)	0.010
No. of received treatment lines before the diagnosis of OM-CRPC			1.10 (0.63–1.76)	0.76
No. of progressive sites			1.48 (0.75–2.72)	0.23
Type of ARAT ^†^	Abiraterone vs. Enzalutamide		1.16 (0.45–3.09)	0.76

OM-CPRC ^#^, oligometastatic castration-resistant prostate cancer; PSDT ^†^, progressive site-directed therapy; PSA *, prostate-specific antigen; PSA-DT ^‡^, PSA-doubling time; ARAT ^†^, androgen receptor-axis-targeted drug; HR ^††^, hazard ratio; CI ^§^, confidence interval.

**Table 3 cancers-14-00567-t003:** Cox hazard model to identify factors predictive of overall survival.

Variable		N	Univariate	
	Category	Patients	HR ^††^ (95% CI ^§^)	*p* Value
Age at the diagnosis of OM-CRPC ^#^			1.05 (0.92–1.21)	0.43
PSA * at the diagnosis of OM-CRPC			1.03 (0.94–1.11)	0.46
PSA-DT ^‡^ at the diagnosis of OM-CRPC			1.10 (0.86–1.31)	0.37
History of definitive therapy for prostate cancer	no vs. yes	13/15	0.42 (0.08–1.93)	0.26
PSDT ^†^	no vs. yes	13/15	0.11 (0.01–0.67)	0.014
No. of received treatment lines before the diagnosis of OM-CRPC			1.12 (0.47–2.46)	0.79
No. of progressive sites			2.15 (0.69–6.46)	0.16
Type of ARAT ^†^	Abiraterone vs. Enzalutamide		0.73 (0.13–3.93)	0.69

OM-CPRC ^#^, oligometastatic castration-resistant prostate cancer; PSDT ^†^, progressive site-directed therapy; PSA *, prostate-specific antigen; PSA-DT ^‡^, PSA-doubling time; ARAT ^†^, androgen receptor-axis-targeted drug; HR ^††^, hazard ratio; CI ^§^, confidence interval.

## Data Availability

Data sharing not applicable.

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
