# Peer review of "Impact of Progressive Site-Directed Therapy in Oligometastatic Castration-Resistant Prostate Cancer on Subsequent Treatment Response"

_cancers, 2022, doi:10.3390/cancers14030567_

Round 1

Reviewer 1 Report

I agree with the authors claim that this is the “first study to show that PSDT [Progressive Site Directed Therapy] has a positive impact on the therapeutic response to subsequent ARAT [Androgen Receptor Axis Targeted] drug and the OS [Overall Survival] in OM-CRPC [Oligo Metastatic-Castrate Resistant Prostate Cancer]”.  Although there are obvious indications that this would be the case from both our knowledge of tumor heterogeneity and similar findings in other cancers, I think that this small retrospective study will be of interest to the readers of Cancers and will stimulate further research.

Author Response

Thank you for understanding the significance of this research.

Reviewer 2 Report

In the current manuscript, the authors performed a retrospective analysis to describe the impact of progressive site-directed therapy (PSDT) on oligometastatic castration-resistant prostate cancer (OM-CRPC) patients who subsequently underwent androgen receptor axis-targeted (ARAT) drugs treatment. Authors mentioned the patients treated with PSDT showed a significant decrease in PSA levels, longer PSA progression-free survival, and longer OS compared to the group of patients who did not receive PSDT. Finally, authors concluded the PSDT may improve the efficacy of subsequent ARAT and OS in OM-CRPC patients. The manuscript is well written and the statistical methods used for the analysis are logical. Though some more details in the introduction and discussion can be added, especially detailing PSDT. The study address an important issue to improve the treatment of OM-CRPC, it has some limitations which authors acknowledged and mentioned in the manuscript.

  1. The authors should mention if the patients included in the analysis and the outcome of the treatment have been published previously, if yes then authors should tabulate and refer to those studies.
  2. Authors mentioned that the PSDT was performed with radiotherapy at all progressing lesions; Prostate/lymph node and bone. The most frequent site for PSDT was bone, can authors comment (or mention in discussion) if the metastatic site dictates the efficacy of PSDT and subsequent hormonal therapy.
  3. Though it was mentioned in the manuscript that ~73% of PSDT group patients had a history of definitive therapy for prostate cancer, how and how much it can affect the subsequent course of treatment.     

Reviewer 3 Report

General comment

This is a retrospective study to assess the influence of PSTD on the efficacy of subsequent ARAT therapy in OMCRPC. It has several limitations, but the results include some important information in this field. The reviewer suggests some issues to be clarified for the accurate interpretation of this study.

Specific comments

  1. L116: Even though 54 patients were included in this study, 26 of them who were not treated with ARAT as the 1st line therapy were excluded from this study. But the exclusion criteria were mentioned in the methods. Thus, the 26 patients should be excluded in the methods and the number of patients in this study should be 28.
  2. Table 1: Even though the difference was not statistically significant, possibly caused by the limited number of subjects, only non-PSDT group included patients with visceral metastasis. It should be mentioned in the limitation.
  3. L153: The median PSA-PFS from diagnosis of OM-CRPC to PSA progression to ARAT was longer in PSDT group than in non-PSDT group. But there should be a lead time bias because ARAT was administrated after PSDT in the PSDT group.
  4. L169: The description “Similarly, … with OS (Table 3).” was duplicated.
